# Prognostic Values of Combined Ratios of White Blood Cells in Glioblastoma: A Retrospective Study

**DOI:** 10.3390/jcm11123397

**Published:** 2022-06-13

**Authors:** Pawel Jarmuzek, Marcin Kot, Piotr Defort, Jakub Stawicki, Julia Komorzycka, Karol Nowak, Anna Tylutka, Agnieszka Zembron-Lacny

**Affiliations:** 1Neurosurgery Center University Hospital, Collegium Medicum University of Zielona Gora, 28 Zyty Str., 65-417 Zielona Gora, Poland; p.jarmuzek@cm.uz.zgora.pl (P.J.); m.kot@cm.uz.zgora.pl (M.K.); stawicki.jakub@gmail.com (J.S.); 2Student Research Group, Collegium Medicum University of Zielona Gora, 28 Zyty Str., 65-417 Zielona Gora, Poland; jmkomorzycka@gmail.com (J.K.); nowakkarol80@gmail.com (K.N.); 3Department of Applied and Clinical Physiology, Collegium Medicum University of Zielona Gora, 28 Zyty Str., 65-417 Zielona Gora, Poland; a.tylutka@cm.uz.zgora.pl (A.T.); a.zembron-lacny@cm.uz.zgora.pl (A.Z.-L.)

**Keywords:** age, brain tumour, high-grade glioma, neutrophils, survival analysis

## Abstract

In some malignant tumours, the changes in neutrophil counts in relation to other blood cells are connected with unfavourable prognosis. Nevertheless, the prognostic value of the combinations of the haematological components in glioblastoma (GBM) remains under dispute. The clinical significance of the neutrophil-to-lymphocyte ratio (NLR), systemic immune inflammation index (SII), and systemic inflammation response index (SIRI) was investigated in our study. We retrospectively studied 358 patients (males *n* = 195; females *n* = 163) aged 59.9 ± 13.5 yrs with newly diagnosed glioma and admitted to the Neurosurgery Centre. Routine blood tests and clinical characteristics were recorded within the first hour of hospital admission. The inflammatory variables: NLR, SII and SIRI exceeded the reference values and were significantly elevated in Grade 3 and Grade 4 tumour. The Cox model analysis showed that the age ≥ 63 years, NLR ≥ 4.56 × 10^3^/µL, SII ≥ 2003 × 10^3^/µL and SIRI ≥ 3.03 × 10^3^/µL significantly increased the risk of death in Grade 4 tumour patients. In the inflammatory variables, NLR demonstrated the highest impact on the survival time (HR 1.56; 95% CI 1.145–2.127; *p* = 0.005). In the first Polish study including GBM patients, the age in relation to simple parameters derived from complete blood cell count were found to have prognostic implications in the survival rate.

## 1. Introduction

Glioblastoma (GBM, WHO grade IV) is the most aggressive primary malignant brain tumour in adults, with an average survival time of 12–18 months after initial diagnosis. GBM patients is based on standard treatment, including surgery, chemotherapy, and radiotherapy [1]. The incidence of GBM is much higher in the elderly individuals, while younger people are more likely to develop lower-grade gliomas. The discovery of somatic mutations in the gene encoding isocitrate dehydrogenase-1 (IDH1) initiated a new period of laboratory diagnostics of GBM [2]. The fifth edition of the World Health Organisation Classification of Tumours of the CNS, published in 2021, established new tumour types and subtypes based on profiling genome-wide DNA methylation [3]. However, disadvantages in diagnostic techniques have impeded their widespread application [4]. Therefore, it has become extremely urgent to find additional markers to predict the outcomes in patients with glioma that are easy to assay. It has long been known that chronic inflammation is an adverse factor that promotes both tumour initiation and progression, and markers of inflammation increase the proliferation and survival of tumour cells and increases the blood supply to tumours [5,6]. The increase in the number of neutrophils, lymphocytes, macrophages or platelets in the peripheral blood is related to the inflammatory response. Due to the strong overproduction of granulocyte colony factor (G-CSF) by neoplastic cells, severe neutrophilia and lymphopenia are observed in patients with gliomas. G-CSF is well recognised as a potent regulator of neutrophil production [7].

Routine blood tests to assess inflammatory processes are often useful in the early diagnosis of several diseases as well as in clinical prognosis in traumatic intracerebral haemorrhage [8]. The assessment of the basic parameters in the peripheral blood count, which is both inexpensive and easy to perform, provides us with information about changes in the numbers of lymphocytes, neutrophils, monocytes and platelets. Recently, some studies have found that the combinations of the haematological components, such as the neutrophil-to-lymphocyte ratio (NLR), platelet-to-lymphocyte ratio (PLR), the lymphocyte-to-monocyte ratio (LMR), systemic immune inflammation index (SII), fibrinogen to albumin (FAR) and C-reactive to albumin (CAR) were effective prognostic indicators in patients with a variety of cancers [9,10,11,12,13,14,15,16]. Particular attention was also paid to the role of platelets in the glioblastoma growth. Whether tumour-secreted inflammatory mediators are the underlying mechanism for tumour-platelet interaction, remains to be elucidated [17]. Haematological markers compared to molecular prognostic markers, such as IDH1 mutation [18], mRNA-profiles [19] and NMR spectroscopy-based metabolomics of blood cells [20] can be used to assess the prognosis of GBM patients in order to guide decisions on further therapeutic procedures. Recently, some meta-analyses [4,21,22,23] demonstrated that a high NLR could be considered a high-risk prognostic factor in GMB, and adjustment in chemotherapy should be recommended for high-risk patients. Therefore, we aimed to assess the prognostic values of combined ratios of blood cells in newly diagnosed glioblastoma and to demonstrate the diagnostic usefulness of the systemic inflammatory variables in prediction of survival time as well as to assess of the impact that the inflammation has on the tumour progress.

## 2. Materials and Methods

### 2.1. Study Population

A retrospective study was carried out in *n* = 358 patients (males *n* = 195 and females *n* = 163) with newly diagnosed glioma who had undergone an operation in Neurosurgery Centre University Hospital in Zielona Gora between August 2004 and May 2021 (Table 1). The overall survival was defined as the interval between the diagnosis and death. For the patients who had not died prior to the last follow-up, the overall survival was censored at the date of the last follow-up. All patients underwent a craniotomy on GBM with either total or subtotal resection. The pathological diagnoses were based on the classification of CNS tumours [2]. The following exclusion criteria were used: biopsy only, age below 18 years, no definite diagnosis, incomplete baseline clinical data, adjuvant therapy like chemotherapy or radiotherapy received before operation, malnutrition and perioperative mortality (survival < 20 days). For 57 patients with the Grade 4 tumour, time of death was not confirmed by (Polish) National Cancer Registry. Eventually, 178 patients with the Grade 4 tumour (males *n* = 91 aged 60.1 ± 13.0 years and females *n* = 87 aged 63.9 ± 11.2 years) and median survival rate of 217 days (the range from 20 to 2946) were included in the survival analysis and regression models. Importantly, every patient diagnosed with a primary brain tumour and enrolled in our study had a very recent diagnosis with no prior specific treatment, including glucocorticoids. The bioethics commission in Zielona Góra approved the research protocol (No. 02/131/2020) in accordance with the Helsinki Declaration.

### 2.2. Clinical Assessment

Medical records were reviewed, and the following clinical data were collected: gender, age at operation, locations and hemisphere of tumours, pathological diagnoses, and some biomarkers. Ki-67 proliferation index was expressed as the percentage of cells with Ki-67-positive immunostained nuclei using the Ventana BenchMark GX (Ventana Medical Systems Inc., Tucson, AZ, USA). The expression of Ki-67 was categorised into two groups: low and intermediate (Ki-67 < 30%), and high (Ki-67 ≥ 30%) according to the recommendations by Chen et al. [24]. The date on postoperative adjuvant therapies and survival time were collected through documentation analysis.

### 2.3. Blood Cells and Inflammatory Variables

Blood samples were collected once preoperatively in a volume of 9 mL for laboratory tests using S-Monovette-EDTA K2 tubes (Sarsted AG & Co. KG, Nümbrecht, Germany) within one hour of admission to hospital, and blood counts were immediately analysed. For the other biochemical analyses, blood samples were centrifuged at 3000 rpm for 10 min, and plasma (3 mL) were stored at −80 °C for further study. Haematological parameters including total white blood cell count (WBC), platelet count (PLT) and differential WBC were determined by Sysmex XN-1000 (Sysmex Europe Gmbh, Norderstedt, Germany). The complete blood count and biochemical analysis was carried out in Central Laboratory of the University Hospital in Zielona Gora. The neutrophil-to-lymphocyte ratio (NLR × 10^3^/µL), the platelet-to-lymphocyte ratio (PLR × 10^3^/µL), the lymphocyte-to-monocyte ratio (LMR × 10^3^/µL), the systemic immune inflammation index (SII × 10^3^/µL = Platelets × Neutrophils/Lymphocytes) and the systemic inflammation response index (SIRI × 10^3^/µL = Monocytes × Neutrophils/Lymphocytes) were calculated and compared to reference values according to Luo et al. [25] and Qi et al. [26].

### 2.4. Statistical Analysis

Statistical analyses were performed using the RStudio, Version 4.1.2 (RStudio, Boston, MA, USA) [27]. The variables were reported as mean values ± standard deviation (SD) or median with interquartile range (iqr). The statistical significance between groups was compared by Kruskal–Wallis rank sum test. The predictive values of variables were evaluated by measuring the area under the receiver operating characteristic curve (ROC curve). The optimal threshold value for clinical stratification (cut-off value) was obtained by calculating the Youden index. Survival curves were plotted by using the Kaplan–Meier method. Cox proportional hazards regression (HR) was performed for both univariate and multivariate analyses. Statistical significance was set at *p*-value < 0.05.

## 3. Results

### 3.1. Study Population

Among 358 study patients 54.5% were males aged 57.4 ± 14.1 years and 45.5% were females aged 59.6 ± 13.6 years. GBM was mostly located in the supratentorial region (frontal, temporal parietal, and occipital lobes), with the highest incidence in the frontal lobe and multiple lobes. Ki-67 ≥ 30% was recorded at 33.6% for all WHO grades and increased to 44.7% in Grade 4 tumour. Patients with high-grade glioma constituted above 50% of the analysed group with a median survival of 217 days and median age of 63 years. No gender differences in the overall survival were observed in Grade 4 tumour (males 327 ± 447 days and females 344 ± 465 days, *p* = 0.806).

### 3.2. Blood Cells and Inflammatory Variables

The WBC were found to fall within the referential ranges and did not differ significantly between groups (*p* = 0.205). Among immune cells, neutrophils showed the greatest changes especially in patients with Grade 3 and Grade 4 tumours. The counts of lymphocytes, monocytes and platelets did not exhibit significant changes compared to reference levels or Grade 1 group (Table 2). NRL, SII and SIRI exceeded the reference values proposed by Luo et al. [21] and were significantly elevated in Grades 2–4 tumour groups. PLR and LMR were found to fall within the referential ranges and did not differ significantly between groups (Table 3).

The results of the ROC analysis of age and inflammatory variables i.e., NLR, SII and SIRI ranged between 0.6 and 0.7, indicating a potential diagnostic value for clinical prognosis for patients with high-grade glioma. The optimal threshold values corresponded to 63 years for age, 4.56 × 10^3^/µL for NLR, 2300 × 10^3^/µL for SII and 3.03 × 10^3^/µL for SIRI (Table 4). Kaplan–Meyer survival curves showed that elder patients (≥63 years) and with NLR, SII or SIRI higher than the optimal threshold value had a significantly higher risk of shorter overall survival time (Figure 1a–d). With regard to the age, the survival probability decreased by 50% in patients aged ≥ 63 years (Figure 1a).

The univariate Cox model analysis confirmed that the age, NLR, SII and SIRI above the optimal threshold values significantly increased the risk of death (Table 5). Among the analysed inflammatory variables, NLR demonstrated the highest impact on the survival rate (HR 1.56, 95% CI 1.145–2.127, *p* = 0.005).

## 4. Discussion

The incidence of glioblastoma increases with age, and the median diagnosis is 64 years. The peak incidence is between the ages of 75 and 84, and after the age of 85, the risk is reduced [28]. With a growing and ageing population, the number of cases is expected to increase. Our retrospective study confirms that the survival probability decreases considerably faster in older than young patients with high-grade glioma. Kaplan–Meier survival curves using cut-off values obtained from ROC curves showed a significantly higher risk of death in patients aged 63–90 than in those aged 23–63 years (Figure 1a). Poorer survival in the older age group has already been reported and attributed to a multitude of age-associated changes including the dysregulation of the immune system. Gan et al. [16] indicated a high NLR as an unfavourable predictor of prognosis for elderly patients with high-grade glioma. We observed that patients aged ≥ 63 years (cut-off value) with NRL ≥ 4.56 × 10^3^/µL ran a significantly higher risk of shorter overall survival time. One of the features of immunoageing is a chronic and systemic low-grade inflammation [29]. The inflammation that often accompanies the elderly is also associated with the development of all stages of tumours, both transformation and metastasis [30]. The tumour-derived molecules can condition the bone marrow to increase the release of neutrophil precursors, such as myelocytes and promyelocytes, thereby leading to an increase in circulating immune cells [31]. The greatest changes in neutrophil counts were observed especially in Grade 3 and 4 tumour patients (Table 2). The function of neutrophils remains controversial as they were shown to have both tumour promoting and limiting properties [32]. Reduced lymphocytes activity and T-cell responses to suppress the immune system may enhance tumour progression and metastasis [33,34]. Neutrophils are also involved in tumour development and the formation of a microinflammatory environment, thereby contributing to tumour growth and angiogenesis [32]. Moreover, neutrophils belong to the main cytokine sources, such as interleukin 6, hepatocyte growth factor, transforming growth factor-β, interleukin 8, and matrix metalloproteinases and they are the factors which play an important role in different stages of tumour development [30,31,32]. Neutrophils release the factors of tumour-related angiogenesis, such as vascular endothelial growth factor, angiopoietin-1 and fibroblast growth factor-2 [35]. A growing body of evidence indicates that the neutrophil count is positively related to GBM grade and is an early predictor of tumour progression in patients with glioblastoma [36]. Lymphocytes are responsible for immune surveillance, and they were reported to be protective prognostic factors for cancer patients [37]. Changes in the T-cell population are associated with immune disorders. It has been observed that a reduction in the number of CD4+ lymphocytes in patients with breast and cervical cancer may be associated with faster tumour growth and lymph node infiltration, but, on the other hand, high levels of CD4+ T lymphocytes can reduce the risk of tumour. Reduction in the CD3+, CD4+, CD8+ and CD56+ T cell population in patients with advanced neoplasms reduces the lymphocyte-dependent antitumor cellular response which results in a worse prognosis [38]. Ding et al. [39] also showed that circulating lymphocytes show a cytotoxic effect, which contributes to the inhibition of tumour proliferation and metastasis. Tumour-activated macrophages can promote tumour growth, invasion and migration. They also induce the apoptosis of CD8+ T cells [40], affect tumour angiogenesis and can be associated with a poor prognosis [41]. By interacting with neoplastic cells in the tumour microenvironment through receptors or downstream effectors, activated platelets can promote proliferation, growth and cell survival [17,42,43]. Our study demonstrated significant changes only in neutrophil count; whereas lymphocytes, monocytes and platelets did not exhibit significant changes compared to reference levels (Table 2). The lymphopenia and thrombocytosis are frequent events during GBM disease progression observed after chemoradiation therapy or adjuvant temozolomide [44,45]. However, chemotherapy or radiotherapy received before the operation were included in the exclusion criteria in our study. Therefore, NLR, SII and SIRI, which are based on neutrophil count, could be taken advantage of to assess pro-tumour immune status, and to predict the survival time in newly diagnosed GBM patients.

Kaplan–Meier survival curves using cut-off values showed that increased levels of NLR, SII and SIRI in Grade 4 tumour patients were significantly associated with the survival rate (Figure 1b–d). However, after multivariate Cox regression analysis, only the NLR remained significantly associated with overall survival. A poorer prognosis was observed in the patients with NLR ≥ 4.56 × 10^3^/µL when compared with the patients with NLR < 4.56 × 10^3^/µL. We also evaluated the PLR and LMR, but neither of these variables correlated with the overall survival. NLR showed the greatest potential in the assessment of survival rate in GBM patients. Meta-analysis by Wang et al. [22] indicated that NLR compared to other inflammatory markers was the independent index for prediction of the overall survival in gliomas, and its preoperative assessment can help to evaluate disease progression, optimise treatment and follow-up in glioma patients. Inflammation markers such as the acute phase positive protein CRP appear insufficient for preoperative prognostic assessment. According to Schnaider et al. [46], the ineffectiveness of CRP may result from a lack of specificity as well as to the possible correction prior to elective surgery related to the potential infection and glucocorticoids administration in GBM patients.

The neutrophilia inhibits the immune system response by suppressing the cytolytic activity of cytotoxic T cells and natural killer cells, which is expressed in further changes in NLR [47,48]. The available studies have shown that neutrophils with intrinsic anti-tumour activity are recruited to the tumour where they are reprogrammed to an immunosuppressive pro-tumour phenotype. The tumour-associated neutrophils (TAN) are capable of supporting tumour progression by promoting the angiogenic switch, by stimulating tumour cell motility, migration and invasion, and by modulating other immune cells as part of the ‘immunosuppressive switch’ [49]. However, neutrophil reprogramming depends on the exposure to various inflammatory mediators found in the tumour microenvironment. For example, the presence of transforming growth factor-β (TGFβ) has been demonstrated to promote a pro-tumour phenotype, whereas the presence of interferon-β or the inhibition of TGFβ signalling has been found to result in TAN of an anti-tumour phenotype [49,50]. GBM has been shown to exhibit the highest neutrophil infiltration among all gliomas, with increased neutrophils recruitment being associated with promoted tumour progression and resistance to treatment [51]. Our study identified the new immune-related prognostic peripheral signals, which are associated with increased inflammation, immune infiltration and activation with a shorter overall survival in GBM patients. Between low-grade (1st and 2nd grade) and high-grade glioma (3rd and 4th grade) the differences were significant mainly with regard to neutrophils and NLR.

Inflammatory indices SII and SIRI above the optimal threshold values significantly increased the risk of death (Figure 1c,d). Taking into account the similarities between the formulas for SII and SIRI, the compatibility between these two inflammatory indices is not surprising, as both indicators have been shown to enhance tumour growth, immune escape and cell survival [51]. By means of multivariate Cox analysis, Topkan et al. [52] demonstrated a significant alliance between a low SIRI and longer progression-free and overall survival durations. The authors indicated SIRI as a novel and independent predictor of survival outcomes in newly diagnosed GBM patients intended to undergo postoperative Stupp protocol. Our study focused on the preoperative period and demonstrated significant changes in the counts of neutrophils and insignificant differences in monocyte and platelet counts. Therefore, we adhere to the concept of a crucial role of neutrophils in GBM progression and/or survival probability for Grade 4 tumour patients, which was confirmed by multivariate Cox regression analysis (Table 5). In this respect, a systemic review of 204 meta-analyses has been carried out [23]. These meta-analyses included individual studies which presented neutrophil counts, NLR or tumour-associated neutrophils categorically as either a high or low value in various types of cancers. In total, 29% of the meta-analyses presented strong or highly suggestive evidence for associations between NLR and overall survival [23]. Although NLR holds a clinical promise in its association with poor GBM prognosis, further observations are required to provide evidence whether NLR may aid clinical diagnostics.

Despite many efforts, glioblastoma patients still pose an interdisciplinary challenge in selecting the right treatment [46]. Therefore, an early and reliable prognostication of a patient’s survival, preferably prior to surgery, has a huge relevance. Currently used imaging methods, such as positron emission tomography or magnetic resonance imaging techniques, NMR spectroscopy-based metabolomics are still extremely valuable both in the diagnosis of malignant brain tumours and in the therapeutic evaluation of GBM. Still though, the differentiation between malignant and inflamed brain tissue remains a challenge with imaging protocols [53]. Moreover, GBM differs from other solid tumours in the low to nearly absent frequency of systemically circulating tumour cells, which enhances the argumentation for the remark of NRL based on the neutrophil count.

## 5. Conclusions

In the first Polish study of the kind, we demonstrated that some parameters derived from complete blood cell analysis, mainly NLR, had prognostic implications in the course of glioblastoma, which may enable neurosurgeons to identify patients with poorer prognosis, and augment personalised approach to surgical treatment strategy.

## 6. Limitations

The limitations of this study include its retrospective nature and the relatively small sample size of Grade 1 and 2 tumour patients, which may have impacted the statistical analysis and affect the final results as a consequence.

## Figures and Tables

**Figure 1 jcm-11-03397-f001:**
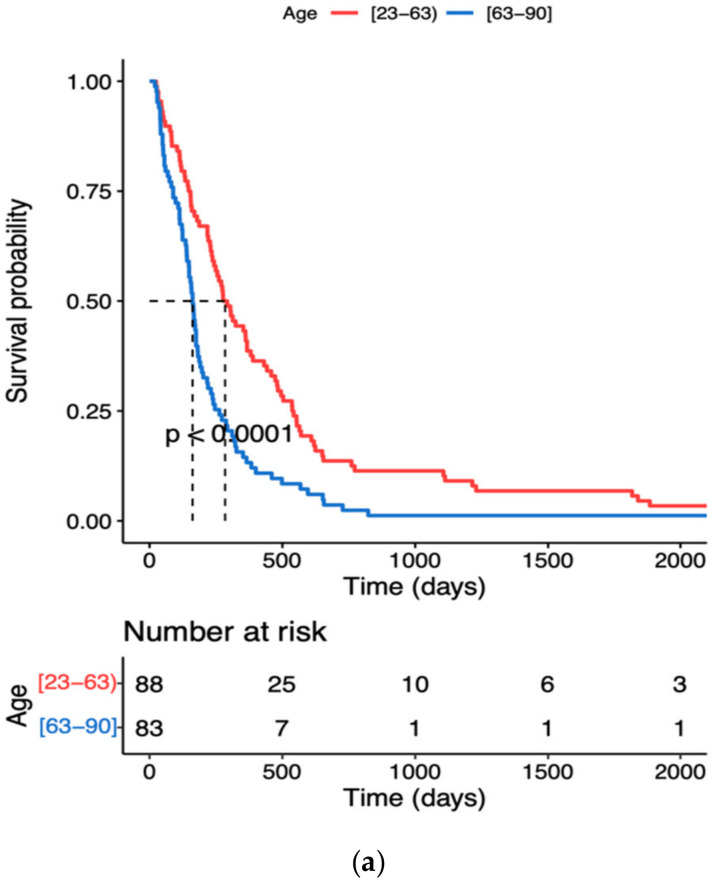
Kaplan–Meier survival curves during hospitalisation of GBM patients with different cut-off values of the age and systemic inflammation markers: (**a**) Age, (**b**) NLR neutrophil to lymphocyte ratio, (**c**) SII systemic immune inflammation index, (**d**) SIRI systemic inflammation response index; the dotted line designates median survival.

**Table 1 jcm-11-03397-t001:** The clinical characteristics of patients with glial tumours (*n* = 358).

		Value (%)
Follow-up period	Mean ± SD (day)	398 ± 575
	Median (range)	211 (1–3702)
Age at operation	Mean ± SD (year)	59.9 ± 13.5
	Median (range)	62.3 (21.9–84.7)
Gender	Males	195 (54.5%)
	Females	163 (45.5%)
Hemisphere	Left	176 (49.2%)
	Right	148 (41.3%)
	Midline or bilateral	34 (9.5%)
Location	Frontal lobe	122 (34.1%)
	Temporal lobe	81 (22.6%)
	Parietal lobe	68 (19.0%)
	Occipital lobe	23 (6.4%)
	Subtentorial location	13 (3.6%)
	Multifocal	51 (14.2%)
Adjuvant therapy	Chemotherapy and radiotherapy	116 (32.4%)
	Chemotherapy or radiotherapy	172 (48.0%)
	None	70 (19.6%)
Ki-67 (all WHO grades)	≥30%	42 (33.6%)
	<30%	83 (66.4%)
Ki-67 (WHO 4th grades)	≥30%	38 (42.7%)
	<30%	51 (57.3%)

Abbreviations: Ki-67, a nuclear protein and a key marker associated with proliferating cancer cells.

**Table 2 jcm-11-03397-t002:** White blood cells and platelets counts (*n* = 358).

Variables	Reference Values	1st Grade *n* = 9		2nd Grade *n* = 32		3rd Grade *n* = 82		4th Grade *n* = 235		*p*-Value
		Mean ± SD	Med (iqr 25–75%)	Mean ± SD	Med (iqr 25–75%)	Mean ± SD	Med (iqr 25–75%)	Mean ± SD	Med (iqr 25–75%)	
WBC (10^3^/µL)	4.0–10.2	9.14 ± 2.17	9.61 (8.15–9.78)	8.07 ± 2.55	7.28 (5.85–9.88)	7.99 ± 2.45	7.63 (5.96–9.93)	8.82 ± 3.07	8.53 (6.33–10.99)	0.205
Neutrophils (10^3^/µL)	2.0–6.9	6.30 ± 2.14	6.19 (5.23–6.30)	6.78 ± 3.93	5.25 (4.21–8.46)	8.31 ± 4.07	7.15 (5.08–11.25)	9.84 ± 6.59	7.85 (6.48–11.62)	<0.001
Lymphocytes (10^3^/µL)	0.6–3.4	2.34 ± 0.71	2.34 (2.12–2.46)	1.61 ± 0.63	1.45 (1.13–2.05)	1.60 ± 0.79	1.47 (1.09–1.88)	1.77 ± 1.25	1.52 (1.06–2.07)	0.718
Monocytes (10^3^/µL)	0.00–0.90	0.59 ± 0.18	0.59 (0.59–0.66)	0.61 ± 0.37	0.50 (0.37–0.80)	0.65 ± 0.29	0.63 (0.46–0.84)	0.71 ± 0.83	0.62 (0.40–0.78)	0.715
Platelets (10^3^/µL)	140–420	283 ± 55	283 (263–310)	262 ± 95	254 (183.0–32.5)	253 ± 73	241 (204.8–292.0)	256 ± 93	245 (197–301)	0.555

Abbreviations: SD, standard deviation; med, median; iqr, interquartile range; WBC, white blood cells.

**Table 3 jcm-11-03397-t003:** Blood cell count-derived inflammation indices (*n* = 358).

Variables	Reference Values	1st Grade *n* = 9		2nd Grade *n* = 32		3rd Grade *n* = 82		4th Grade *n* = 235		*p*-Value
		Mean ± SD	Med (iqr 25%–75%)	Mean ± SD	Med (iqr 25%–75%)	Mean ± SD	Med (iqr 25%–75%)	Mean ± SD	Med (iqr 25%–75%)	
NLR (10^3^/µL)	0.87–4.15	2.96 ± 1.24	2.69 (2.51–3.55)	5.37 ± 5.28	3.36 (2.83–5.66)	7.26 ± 6.10	4.93 (2.95–10.16)	7.68 ± 6.17	5.42 (3.64–10.22)	<0.001
PLR (10^3^/µL)	47–198	138 ± 63	121.0 (98–173)	185 ± 86	172 (123–235)	203 ± 133	168 (126–225)	197 ± 149	159 (105–241)	0.451
LMR (10^3^/µL)	2.45–8.77	4.42 ± 1.69	3.97 (3.97–5.99)	3.36 ± 1.63	2.97 (2.46–3.82	2.86 ± 1.84	2.85 (1.66–3.53)	3.34 ± 3.28	2.76 (1.88–3.69)	0.025
SII (10^3^/µL)	142–808	895 ± 524	763 (513–1222)	1273 ± 931	946 (599–1789)	1829 ±1592	1350 (666–2366)	1964 ± 1800	1319 (776–2548)	0.032
SIRI (10^3^/µL)	0.41–1.42	1.80 ± 1.18	1.59 (0.84–1.59	3.34 ± 5.94	1.82 (1.23–3.48)	4.27 ± 3.51	3.47 (1.55–6.29)	5.64 ± 9.75	3.16 (1.90–5.44)	0.001

Abbreviations: SD, standard deviation; med, median; iqr, interquartile range; WBC, white blood cells; NLR, neutrophil/lymphocyte ratio; PLR, platelet/lymphocyte ratio; LMR, lymphocyte/monocyte ratio; SII, systemic immune inflammation index; SIRI, systemic inflammation response index.

**Table 4 jcm-11-03397-t004:** The statistical characteristics of the ROC curve for the univariate logistic model for Grade 4 tumour patients (*n* = 178).

Variables	AUC	Cut-Off Value	Sensitivity (%)	Specificity (%)
Age	0.720	63.0	32.9	30.2
NLR	0.601	4.56	38.8	41.9
PLR	0.553	282	11.8	75.6
LMR	0.392	2.48	78.8	47.7
SII	0.597	2003	18.8	61.6
SIRI	0.616	3.03	29.4	45.3

Abbreviations: AUC, the area under the curve; cut-off value, the optimal threshold value for clinical stratification.

**Table 5 jcm-11-03397-t005:** Univariate and multivariate Cox model analysis for Grade 4 tumour patients (*n* = 178).

Variables	Univariate			Multivariate		
	HR	95% CI	*p*-Value	HR	95% CI	*p*-Value
Age	1.87	1.367–2.549	<0.0001	1.03	0.967–1.019	<0.0001
NLR	1.56	1.145–2.127	0.005	1.11	0.904–1.025	0.011
SII	1.44	1.030–2.024	0.033	0.99	0.999–1.000	0.074
SIRI	1.50	1.104–2.053	0.0097	1.01	0.989–1.032	0.338

Abbreviations: HR, hazard ratio; 95% CI, confidence interval for the true population value of HR.

## Data Availability

The data used to support the findings of this study are available from the corresponding author upon request.

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
