# Peer review of "Prognostic Values of Combined Ratios of White Blood Cells in Glioblastoma: A Retrospective Study"

_jcm, 2022, doi:10.3390/jcm11123397_

Round 1
Reviewer 1 Report
The manuscript by Jarmuzek and colleagues describes a retrospective study on the prognostic value of blood cell counts in glioma affected patients.
The novelty of the results is quite weak since, as mentioned by the authors in the Discussion, the importance of NLR as main prognostic factor for survival has been already shown in a wide meta-analysis by Wang et al., (2020). Nevertheless, the manuscript is well written and addresses a very important issue.
The title should be changed indicating the main finding. In the present form it is misleading because the mention of platelets seems to indicate they also have a prognostic meaning together with white blood cells.
In the Discussion, the authors should highlight the novelty of their results in comparison with data already present in literature.
Author Response
Response to Review 1
We greatly appreciate your time and effort dedicated to providing feedback on our manuscript and we are grateful for the insightful comments on and valuable improvements to our paper. All the suggestions helped us to evaluate our outcomes even more precisely in order to deliver improved, high quality scientific manuscript which we hope will now meet the high standards of Journal of Clinical Medicine.
Comments and Suggestions for Authors
The manuscript by Jarmuzek and colleagues describes a retrospective study on the prognostic value of blood cell counts in glioma affected patients. The novelty of the results is quite weak since, as mentioned by the authors in the Discussion, the importance of NLR as main prognostic factor for survival has been already shown in a wide meta-analysis by Wang et al., (2020). Nevertheless, the manuscript is well written and addresses a very important issue.
The title should be changed indicating the main finding. In the present form it is misleading because the mention of platelets seems to indicate they also have a prognostic meaning together with white blood cells.
Following the Reviewer’s suggestion, the title has been corrected.
In the Discussion, the authors should highlight the novelty of their results in comparison with data already present in literature.
The Discussion section has been completed according to the Reviewer’s comment.
Following the Reviewer’s suggestion, the manuscript has been revised by professional proofreader.

Reviewer 2 Report
The retrospective study of blood components of glioblastoma patients is very interesting but the authors needs clarify following comments.
1)Material methods should include the details of blood draw, how many times blood drawn during the study course, once before surgery? how much drawn, how the process of separation done. How did you do normalization to compare.
2) Explain why healthy controls are not recruited in the study plan?
3) Survival benefit for NLR greater than 4.56 looks not so great in figure but numbers look very significant. Please clarify. Average how many months of survival benefit will be obtained with these NLR values?
4) Introduction need to include more relevant studies, please cite following papers in your study
i) Tumor–platelet interactions: Glioblastoma growth is accompanied by increasing platelet counts. Clin Neurol Neurosurg. 2008 Apr;110(4):339-42. doi: 10.1016/j.clineuro.2007.12.008. Epub 2008 Feb 20.
ii) RNA-Seq of Tumor-Educated Platelets Enables Blood-Based Pan-Cancer, Multiclass, and Molecular Pathway Cancer Diagnostics. 2015, Cancer Cell 28, 666–676
iii) NMR Spectroscopy-Based Metabolomics of Platelets to Analyze Brain Tumors. Reports 2021, 4(4), 32
Author Response
Review 2
We greatly appreciate your time and effort dedicated to providing feedback on our manuscript and we are grateful for the insightful comments on and valuable improvements to our paper. All the suggestions helped us to evaluate our outcomes even more precisely in order to deliver improved, high quality scientific manuscript which we hope will now meet the high standards of Journal of Clinical Medicine.
Comments and Suggestions for Authors
The retrospective study of blood components of glioblastoma patients is very interesting but the authors need clarify following comments.
1) Material methods should include the details of blood draw, how many times blood drawn during the study course, once before surgery? how much drawn, how the process of separation done. How did you do normalization to compare.
Thank you for this comment. The following sentence has been revised and it now reads as follows: Blood samples were collected once preoperatively for laboratory tests within one hour of admission to hospital, and were immediately analysed. For the other biochemical analyses, blood samples were centrifuged at 3000 rpm for 10 min, and plasma were stored at −80 ◦C for further study.
2) Explain why healthy controls are not recruited in the study plan?
Our retrospective long-term follow-up study included patients admitted to hospital between August 2004 and May 2021, and the same analyzer Sysmex was used to analyze the collected blood samples as well. This was important for the comparison which was made with regard to the reference values indicated/described by Luo H. et al. [Clin Lab 2019], and also for the comparison of the 1st, 2nd, 3rd and 4th grade glioma.
3) Survival benefit for NLR greater than 4.56 looks not so great in figure but numbers look very significant. Please clarify. Average how many months of survival benefit will be obtained with these NLR values?
Thank you for pointing this out. Patients were divided into two groups according to the cut-off value and for the NLR the value was 4.56. „ROCit” package was used to calculate the cut-off value and then based on this, Kaplan-Meier figures were prepared. We can assure you that all the figures were designed using Rstudio software. The following packages from Rsystem were used to create all the figures: („survival”), („survminer”), („lubridate”) and („ggplot”).
As for the patients' survival, we analyzed them in terms of the number of days survived. As suggested by the Reviewer, we have calculated the mean duration of survival in months, and for the 1st group, i.e. NLR [0.55-4.56), the mean duration of survival in months reached 14.2 months, while for the second group [4.56-26.72) the mean the survival time in months was 8.2. This confirms our observation mentioned in Discussion section that "A poorer prognosis was observed in the patients with NLR ≥4.56 x 103/µL when compared with the patients with NLR <4.56 x 103/µL."
4) Introduction need to include more relevant studies, please cite following papers in your study.
The suggested studies have been included to Introduction section. References 17,19 and 20.
- i) Tumor–platelet interactions: Glioblastoma growth is accompanied by increasing platelet counts. Clin Neurol Neurosurg. 2008 Apr;110(4):339-42. doi: 10.1016/j.clineuro.2007.12.008. Epub 2008 Feb 20.
- ii) RNA-Seq of Tumor-Educated Platelets Enables Blood-Based Pan-Cancer, Multiclass, and Molecular Pathway Cancer Diagnostics. 2015, Cancer Cell 28, 666–676
iii) NMR Spectroscopy-Based Metabolomics of Platelets to Analyze Brain Tumors. Reports 2021, 4(4), 32
Following the Reviewer’s suggestion, the manuscript has been revised by professional proofreader.

Round 2
Reviewer 1 Report
It is my opinion that the revised version of the manuscript now fullfils the requirements.